# The Fabrication of High-Hardness and Transparent PMMA-Based Composites by an Interface Engineering Strategy

**DOI:** 10.3390/molecules28010304

**Published:** 2022-12-30

**Authors:** Bo Cao, Peng Wu, Wenxiang Zhang, Shumei Liu, Jianqing Zhao

**Affiliations:** 1School of Materials Science and Engineering, South China University of Technology, Guangzhou 510640, China; 2Key Lab Guangdong High Property & Functional Polymer Materials, and Key Laboratory of Polymer Processing Engineering, Ministry of Education, Guangzhou 510640, China

**Keywords:** PMMA-based composites, dispersion, transparent, surface hardness

## Abstract

The high-hardness and transparent PMMA-based composites play a significant role in modern optical devices. However, a well-known paradox is that conventional PMMA-based composites with high loadings of nanoparticles usually possess high surface hardness at the cost of poor transparency and toughness due to the aggregation of nanoparticles. In this work, ideal optical materials (SiO_2_/PMMA composites) with high transparency and high surface hardness are successfully fabricated through the introduction of the flow modifier Si-DPF by conventional melt blending. Si-DPF with low surface energy and high transparency, which is located at the SiO_2_/PMMA interface, and nano-SiO_2_ particles are homogeneously dispersed in the PMMA matrix. As an example, the sample **SiO_2_/PMMA/Si-DPF (30/65/5)** shows outstanding transparency (>87.2% transmittance), high surface hardness (462.2 MPa), and notched impact strength (1.18 kJ/m^2^). Moreover, **SiO_2_/PMMA/Si-DPF (30/65/5)** also presents a low torque value of composite melt (21.7 N⋅m). This work paves a new possibility for the industrial preparation of polymer-based composites with excellent transparency, surface hardness, processability, and toughness.

## 1. Introduction

Poly (methyl methacrylate) (PMMA) has been widely used as an optical material for its high transparency, light weight, and low cost. Especially in recent years, PMMA has been used as display screens, lenses, and LED encapsulation materials [1,2,3]. PMMA has a high surface hardness and still needs to be improved by incorporating sufficient nanoparticles (i.e., SiO_2_ [4,5,6,7,8,9], TiO_2_ [9], ZrO_2_ [10], Al_2_O_3_ [11], and ZnO [12]) for wider applications. For instance, Tseng and coworkers [6] prepared the SiO_2_/PMMA composites (50/50 by weight) and the surface hardness reached 314 MPa, which was much higher than that of PMMA (196 MPa). Obviously, the high loading of nanoparticles is necessary to attain a high surface hardness of the composites.

However, the nanoparticles are easily packed together and inevitably form a large number of agglomerates in the PMMA matrix, which seriously deteriorates the properties of the composites, especially transparency [13,14]. Tadano et al. [7] introduced 9 wt% nano-SiO_2_ particles into PMMA to prepare the SiO_2_/PMMA hybrid films, and the transmittance of the samples was decreased from 91% to 72% due to the formation of SiO_2_ agglomerates. That is to say, the improvement in surface hardness of the PMMA-based composites by incorporating nanoparticles is achieved at the expense of reduced transparency. Moreover, the serious agglomeration of fillers generally results in poor processability and toughness of the composites, which is still urgently needed to be solved. Hence, incorporating nanoparticles into the PMMA matrix does not guarantee the fabrication of PMMA-based composites with high transparency and high surface hardness [15]. Only when nanoparticles do not aggregate and are uniformly dispersed in the PMMA matrix could high transparency and high surface hardness of the composites be simultaneously achieved [16]. In addition, achieving homogeneous dispersion of nanofillers in the polymer matrix is one of the hot research topics in modern material science [17].

PMMA-based composites are generally prepared by techniques including in-suit polymerization, modified polymerization, hot compression, and solution blending. It has been proven to be effective to introduce nanoparticles into PMMA by in-suit polymerization, forming a homogeneous dispersion of nanoparticles. A SiO_2_/PMMA composite (50/50) was synthesized via in-suit polymerization of methyl methacrylate and 2-(methacryloyloxy) ethyl isocyanate-modified SiO_2_ nanoparticles, and the composite simultaneously possessed a high surface hardness (363 MPa) and high transparency (>89% transmittance) [6]. Unfortunately, the application of the aforementioned method in the industry is difficult due to its complexity [18]. If the homogeneous dispersion of nanoparticles in the PMMA matrix is easy to be realized by melt blending, it is very convenient to fabricate ideal optical materials in an industry [19].

Our previous work demonstrated that the silicone/fluorine-functionalized flow modifier (Si-DPF) with low surface energy was located at the two-phase interface in the magnesium hydroxide/linear low-density polyethylene (MH/LLDPE) composites (80/20), and evidently improved the dispersion of MH particles [20]. In this work, the proposed flow modifier Si-DPF is applied to the production of SiO_2_/PMMA composites by a conventional melt processing technique, aiming to fabricate the SiO_2_/PMMA composites with high transparency, high surface hardness, excellent processability, and toughness. Herein, a small amount of Si-DPF (5 wt%) is introduced into SiO_2_/PMMA composites (the weight ratio of nano-SiO_2_ to PMMA is 10:90, 20:80, and 30:70, respectively). It is expected that the introduction of Si-DPF is conducive to the uniform dispersion of nano-SiO_2_ particles in the PMMA matrix, and the transparency of SiO_2_/PMMA composites is retained while the surface hardness is improved. Moreover, the dispersion and toughening mechanisms are explored. The present work provides a strategy for the development of transparent polymer-based nanocomposites for industrial production.

## 2. Results and Discussion

### 2.1. Phase Morphology

Improving the dispersion of nano-SiO_2_ particles in the PMMA matrix is a key to fabricating transparent SiO_2_/PMMA composites [21,22,23]. As we know, the more the nano-SiO_2_ particles are loaded, the more serious the SiO_2_ agglomeration. To display the effect of Si-DPF on the dispersion of highly filled nano-SiO_2_ particles, the morphology of nano-SiO_2_ particles in the samples **SiO_2_/PMMA (30/70)** and **SiO_2_/PMMA/Si-DPF (30/65/5)** is observed by SEM. Obviously, Si-DPF changes the dispersion of nano-SiO_2_ particles. There is a large amount of SiO_2_ agglomerates in the sample **SiO_2_/PMMA (30/70)** (Figure 1(a_1_,a_2_)), whereas nano-SiO_2_ particles are uniformly dispersed in the PMMA matrix for **SiO_2_/PMMA/Si-DPF (30/65/5)** (Figure 1(b_1_,b_2_)), indicating that Si-DPF plays an important role in the dispersion of highly filled nano-SiO_2_ particles.

Furthermore, to obtain the size and dispersion mechanism of nano-SiO_2_ particles, TEM tests are carried out on SiO_2_/PMMA composites, as shown in Figure 2.

Clearly, for the samples without Si-DPF, there exists a large amount of SiO_2_ agglomerates in the PMMA matrix (Figure 2(a_1_–a_3_)), where the average size of SiO_2_ agglomerates is about 430 nm, 450 nm, and 480 nm, respectively. By contrast, for the SiO_2_/PMMA composites with 5 wt% Si-DPF, the nano-SiO_2_ particles exhibit a homogeneous dispersion in the PMMA matrix and almost retain their original size of 20–40 nm without aggregation (Figure 2(b_1_–b_3_),c). Figure 2(d_1_) provides a TEM image of the sample **SiO_2_/PMMA/Si-DPF (30/65/5)**, in which a relatively thinner interface between the nano-SiO_2_ particles and PMMA matrix is observed. The elemental mapping image (EMI) analysis is performed and the results are shown in Figure 2(d_1_–d_3_). Si signals are observed in the nano-SiO_2_ particles, and F signals are observed in the interface, indicating that Si-DPF tends to be located at the interface between the SiO_2_ particles and the PMMA matrix. Both SEM and TEM tests indicate that Si-DPF is able to effectively prevent nano-SiO_2_ particles from aggregating.

### 2.2. Surface Hardness

The surface hardness of SiO_2_/PMMA composites is evaluated by the nanoindentation tests [24]. Figure 3a shows that the PMMA matrix has a surface hardness of 236.4 MPa. The surface hardness of the samples is increased as the loading of nano-SiO_2_ particles, and the surface hardness is increased to 284.2 MPa for **SiO_2_/PMMA (10/90)**, 355.8 MPa for **SiO_2_/PMMA (20/80)**, and 491.2 MPa for **SiO_2_/PMMA (30/70)**. As expected, the surface hardness is slightly decreased for the samples with Si-DPF. The results show that the long molecular chains of silicone in Si-DPF endow the samples with flexibility and reduce their hardness. Figure 3b shows the load-displacement curves by the nano-indenter. The harder sample requires more loading force for the tip to penetrate the same depth of 500 nm from the surface to the interior of the samples. The results in Figure 3a,b demonstrate that the surface hardness of SiO_2_/PMMA composites is obviously increased with the loading of nano-SiO_2_ particles and is easy to be adjusted.

### 2.3. Optical Properties

The transparency of SiO_2_/PMMA composites is crucial for their application as optical materials [25,26]. Figure 4a shows the transmittance spectra of the PMMA matrix, Si-DPF, and SiO_2_/PMMA samples, and the thickness of the samples is 1 mm. As seen, both PMMA and Si-DPF exhibit excellent transparency, which have a 91.1% and 86.8% transmittance at a wavelength of 760 nm, respectively. The transparency of SiO_2_/PMMA samples is decreased as the loading of nano-SiO_2_ particles, which originated from SiO_2_ agglomeration in the PMMA matrix. The aggregation of nanoparticles in a polymer matrix increases the refraction of light and leads to a decrease in transparency. The transmittance is decreased to 80.4% for **SiO_2_/PMMA (10/90)**, 78.6% for **SiO_2_/PMMA (20/80)**, and 77.3% for **SiO_2_/PMMA (30/70)** at a wavelength of 760 nm. The dispersion of nano-SiO_2_ particles is improved by the introduction of Si-DPF and the transparency of SiO_2_/PMMA samples is obviously increased. The transmittance is increased to 86.5% for **SiO_2_/PMMA/Si-DPF (10/85/5)**, 86.6% for **SiO_2_/PMMA/Si-DPF (20/75/5)**, and 87.2% for **SiO_2_/PMMA/Si-DPF (30/65/5)** at a wavelength of 760 nm, indicating that Si-DPF has an obvious advantage in improving the transparency of SiO_2_/PMMA composites. The transparency of the composites could also be observed with the naked eye. The images of SiO_2_/PMMA samples, as well as the PMMA matrix and Si-DPF, are displayed in Figure 4(b_1_–b_8_), which is unable to be distinguished by sight.

Moreover, the haze (*H*) is also used to evaluate the optical property of the composites, which is given by the ratio of the light diffusely scattered (*T*_d_) to the total light transmitted [27].
*T*_t_) [*H* (%) = *T*_d_/*T*_t_ × 100%] (1)

In Figure 4c, PMMA shows that the haze value is 20.01%. For **SiO_2_/PMMA (10/90)**, when light passes through the interface, light-scattering would happen due to the different refractive index between the nano-SiO_2_ particles and the PMMA matrix, showing a higher haze value (37.27%). However, to our surprise, the haze value is decreased as the increase of nano-SiO_2_ particles loading, and the corresponding values are decreased to 27.04% for **SiO_2_/PMMA (20/80)** and 17.66% for **SiO_2_/PMMA (30/70)**. This is attributed to the more serious agglomeration of SiO_2_ particles at the higher loading. The average size of SiO_2_ agglomerates increased and the total area of the interface between the PMMA matrix and SiO_2_ particles decreased. Consequently, the light-scattering is weakened and the haze values of the samples present a downward trend [28,29,30]. With the addition of Si-DPF, the nano-SiO_2_ particles exhibit a homogeneous dispersion in the PMMA matrix, and the total area of the interface between the PMMA matrix and SiO_2_ particles is obviously increased. The haze values are increased to 57.10% for **SiO_2_/PMMA/Si-DPF (10/85/5)**, 38.35% for **SiO_2_/PMMA/Si-DPF (20/75/5)**, and 34.26% for **SiO_2_/PMMA/Si-DPF (30/65/5)**, respectively. The results further confirm that the Si-DPF is effective in improving the dispersion of nano-SiO_2_ particles in the PMMA matrix.

Table 1 summarizes the reported transparency and surface hardness enhancement for PMMA-based composites with various nanoparticles. As seen, few studies on PMMA-based composites have focused on transparency and surface hardness simultaneously. It is noted that the PMMA-based composites with higher nano-SiO_2_ particle contents in this work show higher transparency and higher surface hardness enhancement. This work provides a relatively more efficient and facile method to improve the transparency and surface hardness of PMMA-based composites.

### 2.4. Processability

The processability of the composites is critical for engineering applications, which is evaluated by the torque rheology test. Figure 5 shows the torque vs. time curves for SiO_2_/PMMA composites. As seen, the equilibrium torque of composite melt is increased as the loading of nano-SiO_2_ particles and Si-DPF can decrease the melt torque. For samples **SiO_2_/PMMA (10/90)**, **SiO_2_/PMMA (20/80),** and **SiO_2_/PMMA (30/70)**, the stable torque values are 19.0 N⋅m, 23.1 N⋅m, and 26.3 N⋅m, respectively. With the addition of Si-DPF, the torque values of the samples are decreased, and the corresponding ones are decreased to 15.0 N⋅m for **SiO_2_/PMMA/Si-DPF (10/85/5)**, 17.1 N⋅m for **SiO_2_/PMMA/Si-DPF (20/75/5)**, and 21.7 N⋅m for **SiO_2_/PMMA/Si-DPF (30/65/5)**, respectively, indicating an improvement in processability. The complex viscosity *η** is also used to evaluate the processability of composites, and Figure 5b shows the curves of complex viscosity (*η**) vs. frequency (ω) for the samples. As seen, the *η** of samples are obviously increased with the loading of nano-SiO_2_ particles, and the viscosity values of SiO_2_/PMMA samples are decreased with the addition of Si-DPF. This result is ascribed to the improvement in the dispersion of nano-SiO_2_ particles; that is, Si-DPF tends to be located at the interface, which prevents nano-SiO_2_ particles from aggregation and results in a decrease in the melt viscosity [20].

### 2.5. Toughness

Usually, the toughness of high-filled PMMA is decreased. To assess the toughness of SiO_2_/PMMA composites with and without Si-DPF, the notched impact test of the PMMA matrix and the SiO_2_/PMMA composites is conducted, and the results are shown in Figure 6. As seen, the PMMA matrix presents an impact strength of 1.12 kJ/m^2^. Undoubtedly, the notched impact strength of SiO_2_/PMMA samples is decreased, as the loading of nano-SiO_2_ particles and the strengths are decreased to 1.04 kJ/m^2^ for **SiO_2_/PMMA (10/90)**, 0.99 kJ/m^2^ for **SiO_2_/PMMA (20/80)**, and 0.96 kJ/m^2^ for **SiO_2_/PMMA (30/70)**. The decrease in strength originated from the SiO_2_ agglomerates in the PMMA matrix. By contrast, the notched impact strength of **SiO_2_/PMMA/Si-DPF (10/85/5)**, **SiO_2_/PMMA/Si-DPF (20/75/5),** and **SiO_2_/PMMA/Si-DPF (30/65/5)** are 1.17 kJ/m^2^, 1.21 kJ/m^2^, and 1.18 kJ/m^2^, respectively. These samples exhibit a higher notched impact strength and better toughness compared to the PMMA matrix, showing that Si-DPF has a positive effect on improving the toughness of SiO_2_/PMMA composites. Si-DPF with low surface energy tends to be located at the phase interface, which not only plays the role of dispersing nano-SiO_2_ particles but also transfers the energy. When the samples are impacted by the external force, the energy is transferred to the small aggregates, which act as a stress concentration point to dissipate energy; thus, the toughness is improved.

Nano-SiO_2_ particles aggregate easily due to the high surface energy. Therefore, weakening the interaction between nano-SiO_2_ particles is critical for improving the dispersion of nano-SiO_2_ particles. Si-DPF with low surface energy prefers to locate at the SiO_2_/PMMA interface, forming a protective layer on the SiO_2_ particles, and reducing the SiO_2_ agglomeration. For the highly filled SiO_2_/PMMA composites, the uniformly dispersed particles could help to improve the transparency of the composites while maintaining high surface hardness. As a protective layer, Si-DPF contributes to the relative sliding of nano-SiO_2_ particles under an applied stress during the melt blending stage, reducing the melt viscosity of the samples and thus improving processability.

## 3. Experimental

### 3.1. Main Materials

The commercially available poly (methyl methacrylate) (PMMA, CM-211) was purchased from Chimei Taiwan Co., Ltd., with a melt flow rate of 7 g/10 min (Tainan, Taiwan, China). Silica (nano-SiO_2_, A-200, 2.2 g/cm^3^) was supplied by Shandong Dongyue Silicone Material Co., Ltd., with a specific area of 180–220 m^2^/g and a mean diameter of 10–40 nm (Zibo, Shandong, China). The synthesis of the silicone/fluorine-functionalized flow modifier (Si-DPF) referred to our previous work (CST. 2021, 214, and 108994) [20].

### 3.2. Fabrication of SiO_2_/PMMA Composites

PMMA was dried in a vacuum oven at 80 °C for 12 h. SiO_2_/PMMA composites were prepared by melt blending in a twin-screw extruder at a screw speed of 150 rpm. The extruder was configured with ten heating zones, and the extrusion was carried out at zone temperatures of 165 °C, 170 °C, 170 °C, 175 °C, 175 °C, 175 °C, 180 °C, 180 °C, 185 °C, and 185 °C, respectively. After being granulated and dried, the pellets were injection-molded on an injection-molding machine to obtain different specimens. The temperatures of the injection-molding machine were set in a range from 175 °C to 195 °C. The formulation of SiO_2_/PMMA samples is listed in Table 2.

### 3.3. Analysis and Characterization

Scanning electron microscopy (SEM) images were obtained with a Nova NanoSEM430 scanning electron microscope (FEI, Hillsboro, OR, USA) operating at 20 kV. Samples were fractured in liquid nitrogen and the fracture surfaces were coated with gold before observation. Transmission electron microscopy (TEM) images were obtained with a JEM 2100F transmission electron micrograph (JEOL, Tokyo, Japan) operating at 100 kV. Samples were prepared by cutting the blends into approximately 50 nm slices with an RMC PowerTome. Transmittance spectra of the samples were recorded on a Hitachi U-3900H spectrometer (Hitachi, Tokyo, Japan) in a range from 500 to 800 nm. The haze of the samples was measured by an SGW-820 haze meter (Shanghai INESA Physico-Optical Instrument, Shanghai, China) under standard illuminant D65 and in compliance with ASTM D1003. The surface hardness of the samples was measured on a TTX-NHT^3^ nanoindentation instrument (Anton Paar, Graz, Austria) with a Berkovich diamond tip at room temperature. The test speed was constant at 5 mN/min and the maximum depth was 500 nm. The processability of the samples was measured on a Haake XSS-300 torque rheometer (Haake, Vreden, Germany) at a rotation rate of 50 rpm at 180 °C for 10 min. The complex viscosity of the samples (the sample disks with 25 mm diameter and 1 mm thickness) was carried out on a stress-controlled rheometer (AR-G2, TA Instruments, NC, USA) in a dynamic frequency sweep from 0.1 to 628 rad s^−1^ at a strain of 1% within the linear viscoelastic range at 180 °C. According to the ISO 179-1:2000 standards, the impact strength of the samples was tested on a Zwick5113 impact pendulum machine (ZwickRoell, Ulm, Germany). At least five specimens were tested for each measurement and the average results were reported.

## 4. Conclusions

We successfully achieved the properties of high transparency, high surface hardness, excellent processability, and toughness in high-performance SiO_2_/PMMA composites by using the conventional melt mixing technique, solving the well-known problem that high loading of nano-SiO_2_ particles is easy to aggregate and the problem of SiO_2_ agglomerates deteriorating the transparency and toughness of the composites. The 30 wt% nano-SiO_2_ particles, uniformly dispersed in the PMMA matrix with the help of Si-DPF located at the phase interface, facilitated the building of a perfect composite with high transparency and high surface hardness, compared with SiO_2_/PMMA composites with a serious agglomeration of SiO_2_. In addition, the fabricated SiO_2_/PMMA composites also presented excellent toughness and processability. This work provides a quick and effective strategy for improving the production efficiency of commercially available polymer-based composites with high transparency and high surface hardness, which pave a way for their application as ideal optical materials, such as display screens, lenses, and LED lighting.

## Figures and Tables

**Figure 1 molecules-28-00304-f001:**
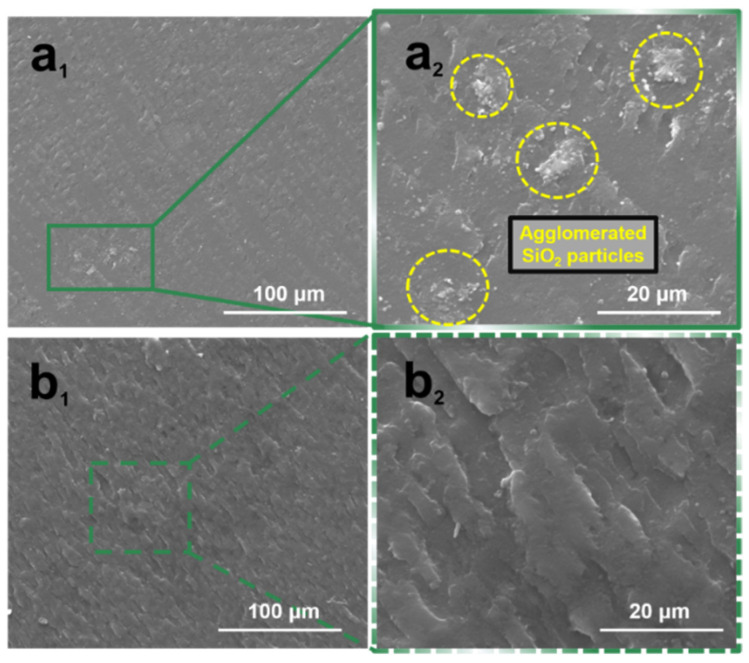
SEM images of SiO_2_/PMMA (30/70) (**a_1_**,**a_2_**) and SiO_2_/PMMA/Si-DPF (30/65/5) (**b_1_**,**b_2_**).

**Figure 2 molecules-28-00304-f002:**
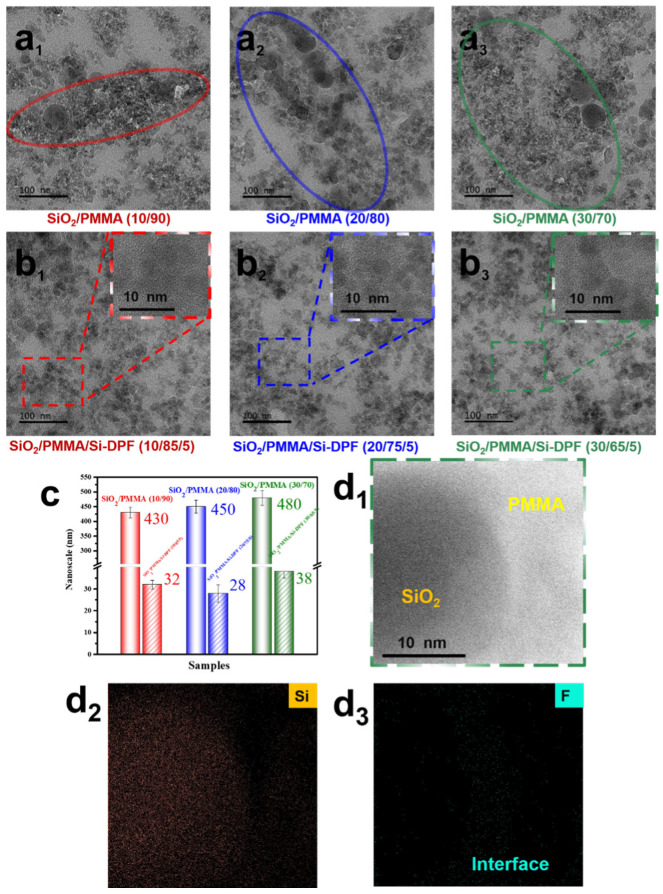
TEM images of SiO_2_/PMMA composites. (**a_1_**) SiO_2_/PMMA (10/90), (**a_2_**) SiO_2_/PMMA (20/80), (**a_3_**) SiO_2_/PMMA (30/70), (**b_1_**) SiO_2_/PMMA/Si-DPF (10/85/5), (**b_2_**) SiO_2_/PMMA/Si-DPF (20/75/5), and (**b_3_**,**d_1_**) SiO_2_/PMMA/Si-DPF (30/65/5). (**c**) SiO_2_ particle size of SiO_2_/PMMA composites. (**d_2_**,**d_3_**) Elemental mapping images of Si and F.

**Figure 3 molecules-28-00304-f003:**
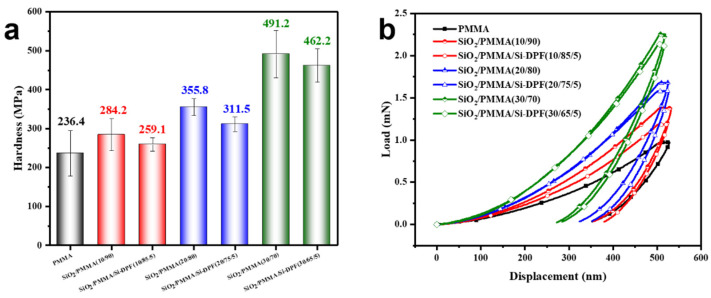
Surface hardness of the PMMA matrix and SiO_2_/PMMA composites. (**a**) Hardness. (**b**) Load-displacement curves.

**Figure 4 molecules-28-00304-f004:**
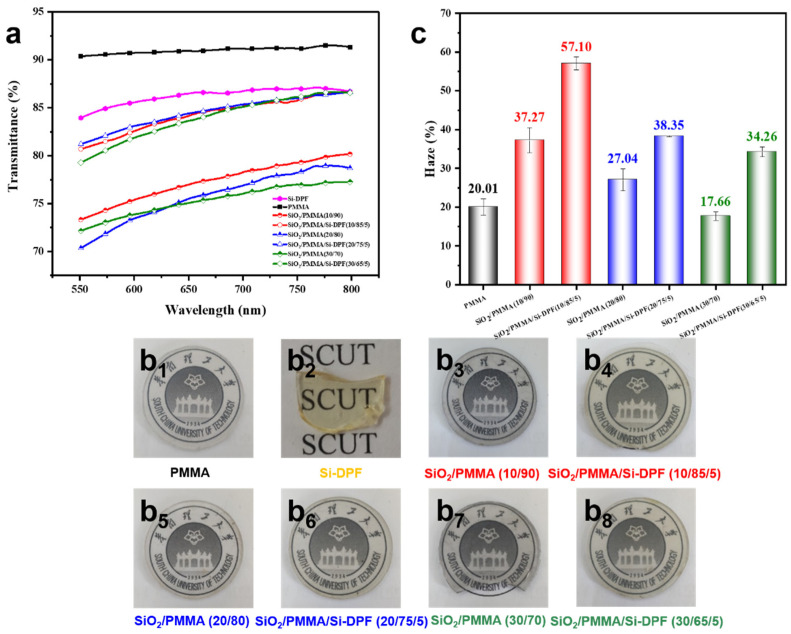
(**a**) Transmittance spectra of PMMA matrix, Si-DPF, and SiO_2_/PMMA samples with a thickness of 1 mm. (**b**) Images of PMMA matrix (**b_1_**), Si-DPF (**b_2_**), and SiO_2_/PMMA samples (**b_3_**–**b_8_**). (**c**) The haze value of the PMMA matrix and SiO_2_/PMMA samples.

**Figure 5 molecules-28-00304-f005:**
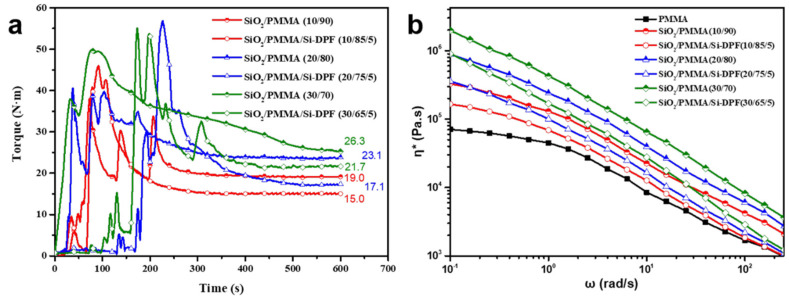
(**a**) Torque rheological analysis of SiO_2_/PMMA composites. (**b**) Complex viscosity versus frequency of samples. Temperature: 180 °C.

**Figure 6 molecules-28-00304-f006:**
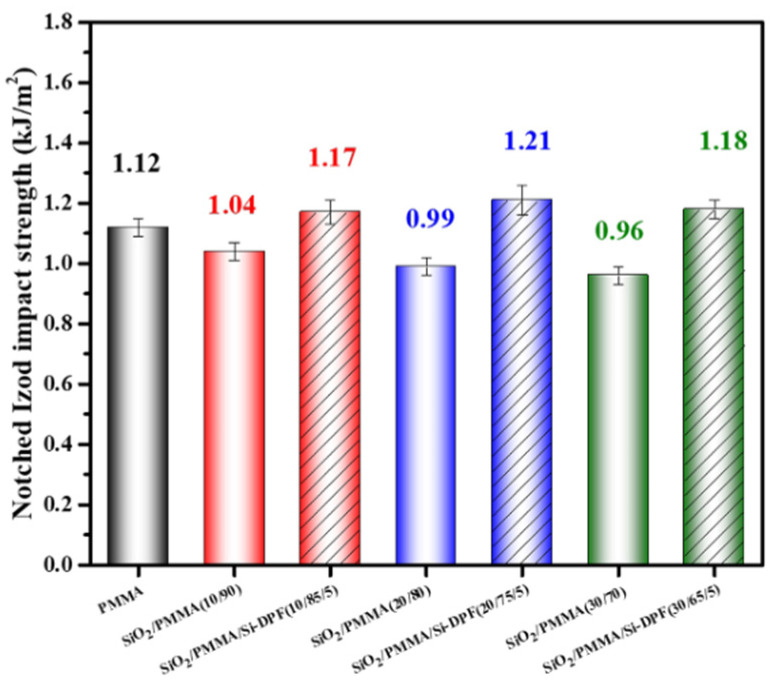
Impact strengths of the PMMA matrix and SiO_2_/PMMA composites.

**Table 1 molecules-28-00304-t001:** A comparison of the transmittance and surface hardness enhancement in PMMA-based composites with various nanoparticles.

Samples	Strategy	Filler Loading [wt%]	Transmittance [%, 760 nm]	Surface Hardness Enhancement [%]	Year [Ref]
SiO_2_/PMMA	In situ polymerization	37.5	90	80.0	2005 [4]
SiO_2_/PMMA		4	78	-	2005 [5]
ZnO/PMMA		5	65	-	2018 [12]
SiO_2_/PMMA		≈3	88	-	2020 [9]
TiO_2_/PMMA		≈3	87	-	2020 [9]
SiO_2_/PMMA	Modified polymerization	50	89.5	85.2	2006 [6]
SiO_2_/PMMA		13.5	90	-	2019 [8]
ZrO_2_/PMMA	Hot compression	1.5	-	12.3	2011 [10]
Al_2_O_3_/PMMA		3	-	29.8	2020 [11]
SiO_2_/PMMA	Solution blending	9.1	72	-	2014 [7]
SiO_2_/PMMA	Melt blending	30	87.2	95.5	This work

**Table 2 molecules-28-00304-t002:** Formulation of various SiO_2_/PMMA samples.

Samples	SiO_2_ (g)	PMMA (g)	Si-DPF (g)
SiO_2_/PMMA (10/90)	10	90	-
SiO_2_/PMMA/Si-DPF (10/85/5)	10	85	5
SiO_2_/PMMA (20/80)	20	80	-
SiO_2_/PMMA/Si-DPF (20/75/5)	20	75	5
SiO_2_/PMMA (30/70)	30	70	-
SiO_2_/PMMA/Si-DPF (30/65/5)	30	65	5

## Data Availability

Not applicable.

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
