# Peer review of "The Fabrication of High-Hardness and Transparent PMMA-Based Composites by an Interface Engineering Strategy"

_molecules, 2022, doi:10.3390/molecules28010304_

Round 1
Reviewer 1 Report
The high-hardness and transparent PMMA-based composites are very promising as optical materials, such as display screens, lenses and LED encapsulation materials. Cao et al report a kind of optical materials with high transparency and high surface hardness by conventional melt blending and by the introduction of the flow modifier Si-DPF with low surface energy and high transparency. I think the manuscript can be accepted after a major reversion. The comments are as below: 1 The experiment shown in Table 1 is quite messy and irregular for too many variables. Furthermore, if there is one variable, the experiment data is no more than 3. 2 Table 2 is very important, but it should be reorganized because it is difficult to find the rule from Table 2. 3 The mechanism for preparation high-hardness and transparent PMMA-based composites should be disccussed.
Reviewer 2 Report
1. Correct the literature review. Describe the references [4-12] in more detail.
Complete the literature review by the method of combating agglomeration of dispersed structures in solid matrices - mechanical dispersion:
Shchegolkov, A.V.; Jang, S.-H.; Shchegolkov, A.V.; Rodionov, Y.V.; Glivenkova, O.A. Multistage Mechanical Activation of Multilayer Carbon Nanotubes in Creation of Electric Heaters with Self-Regulating Temperature. Materials 2021, 14, 4654. https://doi.org/10.3390/ma14164654.2. Describe in more detail the purpose and objectives of scientific work.
2. Clarify the modes of measurement using electron microscopy.
3. Specify the country of the manufacturer and the name of the manufacturers of the research equipment.
4. Complete the conclusion and abstract. Specify in the conclusion in which specific optical devices the developed material can be used.
Round 2
Reviewer 1 Report
I think the manuscript has been modified and it can be accepted for publication in present form.